# Women’s Knowledge about Pharmacological and Non-Pharmacological Methods of Pain Relief in Labor

**DOI:** 10.3390/healthcare11131882

**Published:** 2023-06-29

**Authors:** Jakub Pietrzak, Wioletta Mędrzycka-Dąbrowska, Andrzej Wróbel, Magdalena Emilia Grzybowska

**Affiliations:** 1Department of Obstetrics and Gynecological Nursing, Medical University of Gdansk, Dębinki 7, 80-211 Gdańsk, Poland; 2Department of Anesthesiology Nursing & Intensive Care, Medical University of Gdansk, Dębinki 7, 80-211 Gdańsk, Poland; wioletta.medrzycka@gumed.edu.pl; 3Second Department of Gynecology, Medical University of Lublin, Jaczewskiego 8, 20-954 Lublin, Poland; wrobelandrzej@yahoo.com; 4Department of Gynecology, Obstetrics and Neonatology, Medical University of Gdansk, Smoluchowskiego 17, 80-214 Gdańsk, Poland; magdalena.grzybowska@gumed.edu.pl

**Keywords:** knowledge, labor pain, pharmacological methods, non-pharmacological methods

## Abstract

This study aimed to assess knowledge about non-pharmacological pain-relief methods in labor among women who have given birth at least once. This cross-sectional study was conducted using an online survey among 466 adult women. The minimum sample size was estimated based on the number of labor admissions in the year before the study in Poland. The survey included questions about respondents’ sociodemographics and knowledge of pain-relief methods. The knowledge score was calculated using the sum of correct answers. Non-parametric Spearman’s correlation, Kruskal–Wallis and Wilcoxon variance tests were used. Antenatal classes (313/68.9%) and the Internet (248/54.6%) were the most common sources of knowledge. The most popular pharmacological pain-relief methods included epidural anesthesia (386/85.0%) and nitrous oxide (301/66.3%). Massage and breathing techniques were the most commonly known non-pharmacological methods (405/89.2% and 400/88.1%, respectively). The knowledge score about non-pharmacological methods was significantly higher as compared to the pharmacological methods score (r_c_ = 0.85; *p* < 0.001). Respondents’ age correlated with knowledge about non-pharmacological methods (r_s_ = −0.10_,_
*p* = 0.026) but did not correlate with knowledge about pharmacological methods. Educational level correlated with knowledge about pharmacological (r_s_ = −0.13_,_
*p* = 0.007) and non-pharmacological (r_s_ = 0.14, *p* = 0.003) methods concerning pain relief in labor. No correlation was found between respondents’ knowledge and gravidity, number of vaginal or cesarean deliveries, and hospital referral levels for previous deliveries. Our findings support the need to implement educational programs to increase evidence-based knowledge about pain-relief methods during labor in women.

## 1. Introduction

The majority of women perceive labor as the most arduous and painful ordeal in their life [1,2]. Each woman experiences labor in her own unique way. Labor, although a natural and physiological process, constitutes a physiological but also a psychological challenge for most women [2,3]. The most common emotions that accompany labor include excitement, fear, anxiety and uncertainty [4]. When asked to identify the emotions they feel during labor, the vast majority of the respondents mentioned pain [5]. According to the literature, numerous factors affect the feeling and the experience of labor pain, including previous childbirth experiences, as well as social and cultural factors, i.e., education, religious beliefs, and the circadian rhythm [6,7,8,9]. Pain intensity increases with the progression of labor and depends on the fetal size and passage through the birth canal, as well as maternal threshold levels for pain [7]. Most parturients require pain-relief methods during childbirth. The strategies of labor pain relief include pharmacological (aimed at relieving pain) and non-pharmacological (aimed at helping to deal with pain) interventions [7,8]. Pain-relief methods are associated with a number of advantages as well as disadvantages. Systematic reviews have demonstrated that non-pharmacological methods are inexpensive and easily applicable, help women to become active participants in the decision-making process about whether or not to use pain relief, and are associated with relatively few adverse effects, if any. On the other hand, there are more empirical data which confirm that most pharmacological methods are more effective in relieving labor pain, but they may be associated with adverse effects in the child and/or the mother [10,11]. As far as the criteria for optimal ways of relieving labor pain are concerned, the methods in question should be safe and effective, should not affect the mobility of the woman or progress of labor, and need to match each woman’s preference for the desired mode of delivery [7]. Knowledge about labor pain is vital to assist the women in developing strategies which will limit the use of pharmacological interventions [1,8,9]. The literature reports emphasize that negative emotions during labor may lead to fear of childbirth or even prevent the women from getting pregnant again. Fear of labor pain during the next delivery may also trigger postpartum depression. Pathological fear of childbirth—tokophobia—is found in 6–10% of pregnant women. Tokophobia is a multidimensional phenomenon, stemming from numerous biological, psychological, and social factors [12,13]. According to the literature, over 23% of primiparas describe the pain as ‘unbearable’, 60% as ‘very strong or strong’, and only 9% as ‘bearable’. Labor pain is characterized in a slightly different way by multiparas: 17% describe the pain as ‘unbearable’, 46% as ‘very strong or strong’, and 25% as ‘bearable’ [14,15]. It needs to be emphasized that labor pain relief is a standard component of perinatal care [16]. Studies have shown that knowledge about the methods of labor pain relief and their effectiveness resulted in higher satisfaction with childbirth [2,17,18]. The appropriate use of non-pharmacological pain-relief techniques involves prior knowledge and proper guidance that enables women to use them correctly and effectively to relieve pain and anxiety and control labor [2,18]. Importantly, studies assessing the knowledge of women about the methods of relieving labor pain are scarce, and the available reports found their knowledge to be rather limited [5,18]. According to previous studies, there is a lack of knowledge regarding the need for pain relief during labor, the various types of labor pain-relief methods and their advantages and disadvantages [18].

Therefore, the purpose of our study was to assess the most frequently chosen sources of knowledge about methods of relieving labor pain in an era of unlimited access to various sources of information. Nowadays, the reliability of the information obtained is particularly important. This study aimed to measure the level of women’s knowledge about both non-pharmacological and pharmacological pain-relief methods in labor, and the factors affecting the extent of that knowledge.

## 2. Material and Methods

### 2.1. Study Design

This was a cross-sectional study conducted using an online survey form shared on social media groups only for pregnant and postpartum women. The project was carried out among 466 women in 2021. Participation in the study was anonymous and voluntary. Informed consent was obtained before completing the form.

### 2.2. Participants

The study group included postpartum adult women (>18 years old) who received labor pain relief during previous delivery and consented to complete the online-based survey from. The inclusion criteria were the following: using labor pain-relief methods, adult age, singleton pregnancy and 36 or more weeks pregnancy. Instrumental delivery did not exclude participation in the study. The exclusion criteria were the following: participants under the age of 18, who could not communicate in Polish, women who had never been pregnant, women who were eligible for elective caesarean section and those who refused to participate in the study.

### 2.3. Instrument

The study was conducted using a survey form consisting of two parts. The first part included sociodemographic questions about age, place of residence, level of education, parity, and hospital referral level. There are three levels of perinatal care system in Poland. The hospitals are assigned their referral status (primary, secondary, and tertiary) based on the available level of specialist care, equipment, medical personnel, and resources. The second part included questions about participation in antenatal classes, sources of information about methods of relieving labor pain, both non-pharmacological and pharmacological, and their influence on the course of labor.

The knowledge score was calculated on the basis of the sum of correct answers to the questions in the online survey form. One point was awarded for each correct answer, and 0 points were assigned to each incorrect answer. The points for each of the investigated areas (non-pharmacological methods, pharmacological methods, and basic knowledge about pain relief in labor) were calculated separately. A higher score was indicative of a higher number of correct answers, i.e., greater knowledge about pain relief in labor. The highest possible score was 16 points. A lower score corresponded to fewer correct answers and, consequently, lower knowledge about pain relief in labor. The questions are presented in Table 1.

### 2.4. Data Collection and Research Sample

Data collection took place between April and October 2021. A total of 466 women participated in the survey. Out of those, 12 pregnant women who did not give birth yet were excluded. Therefore, 454 women who gave birth at least once were enrolled into the study. The basis for estimating the minimum size of the sample was the number of patients admitted to labor in the year preceding the study in Poland where the study was conducted (*n* = 355,300) [19]. With a confidence level of 95% a structure ratio of 50% and an assumed maximum error of 5%, the cohort should consist of 385 women. There were no missing data due to the electronic submission of survey responses. The online survey was designed to require respondents to answer all questions to complete the survey. Participants received no feedback regarding correct or incorrect answers after completing the questionnaire.

### 2.5. Outcomes

The primary outcome of the study was the level of patient knowledge about labor pain relief. We investigated the sources of their knowledge, method division, and their impact on the course of labor. Sociodemographic factors were the secondary outcomes of the study. A correlation between the level of knowledge about methods of pain relief in labor and delivery-related variables, and selected sociodemographic variables such as age, education, and place of residence was analyzed. The history of delivery in hospitals with different referral levels (primary, secondary, and tertiary) was also analyzed in relation to the level of knowledge in the study group.

### 2.6. Statistical Analysis

Statistical analysis was conducted using Statistica 13.3 (TIBCO Software Inc., Palo Alto, CA, USA). Non-parametric Spearman’s correlation and H Kruskal–Wallis and Wilcoxon variance tests were used with the following coefficients: effect size, epsilon-square (ε^2^), and matched-pair coefficient (r_c_). Frequency distribution and quantitative data distribution were presented in the first part of the statistical analysis. The choice of the non-parametric tests resulted from non-normal distribution of the quantitative data, assessed with the use of the Kolmogorov–Smirnov test and ordinal measure of the labor-related sociodemographic variables. The α-value of <0.05 was considered statistically significant.

## 3. Results

### 3.1. Participants’ Characteristics

Women between the ages of 31 and 35 constituted the largest group of subjects, almost 50% of the sample, while young women (18–23 years) were the least numerous. The vast majority of the women had higher education (85%). More than half of the study population resided in big towns (>250,000 residents). In the vaginal delivery subpopulation, women who gave birth once constituted the largest group. Only 31% of the women underwent cesarean delivery. More than 3 vaginal deliveries were reported by only 3% of the women, but no cases of >3 cesarean sections were found. The number of deliveries was equally distributed among the different hospital referral levels and ranged from 32% to 34% (Table 2).

### 3.2. Sources of Information and Knowledge about Pain Relief in Labor

Sources of knowledge about pain relief in labor were analyzed (Figure 1). The most commonly listed sources included antenatal classes and the Internet (68.9% and 54.6%, respectively). Notably, the Internet was listed significantly more often as a source of knowledge than a midwife, a physician, or professional literature. Friends are also more often listed as a knowledge source than a physician (19.2%).

The most commonly listed pharmacological methods of pain relief in labor included epidural anesthesia and nitrous oxide (85.0% and 66.3%, respectively). Significantly fewer women (19.6%) were aware of opioids as a pharmacological method of pain relief. As for non-pharmacological methods, massage and breathing techniques were the most commonly reported ways of relieving pain during childbirth (89.2% and 88.1%, respectively). Upright birthing positions (75.5%), immersion in water (71.4%), and physical activity (63.9%) were also frequently mentioned by the respondents. On the other hand, hypnosis, music therapy, or TENS were known to a significantly smaller number of women (34–36%) (Figure 2). A very small percentage of women (0.7%) gave incorrect answers to the questions about pharmacological methods and selected general anesthesia as an example of a pharmacological method of pain relief in labor.

The respondents were asked about the benefits of using non-pharmacological and pharmacological methods of pain relief in labor (Figure 3). Pain reduction was the main benefit listed by the respondents, both in cases of non-pharmacological and pharmacological methods (92.7% and 90.5%, respectively). As far as non-pharmacological methods are concerned, easier delivery, better oxygenation of the fetus, and control over one’s body were reported as benefits. Relaxation was also mentioned as a perceptible benefit of using non-pharmacological methods of pain relief (41.0%). No benefits associated with pain relief using non-pharmacological and pharmacological methods were reported only by a few women (2.6% and 4.6%, respectively). Benefits of using pharmacological methods were reported by 38–58% women and included control over one’s body, easier delivery, and more satisfaction with labor.

Distribution analysis of the abovementioned scores demonstrated non-normal distribution in all measures of knowledge about pain relief in labor (Table 3). Data distribution lacked skewness, both for non-pharmacological and pharmacological methods. Closeness of the median and mean for these values, and similar differences between the minimum value and the first quartile and the maximum value and the third quartile indicate the lack of skewness for the obtained results, which is confirmed by skewness near zero. The level of knowledge about pain-relief methods in labor was distinctly left-skewed. The second- and third-quartile measures of that value were also maximum values. Many respondents (at least half) scored the maximum points in the ‘basic knowledge’ area.

Analysis of variance was used to measure the level of respondent knowledge about non-pharmacological and pharmacological methods of pain relief. Wilcoxon variance test for matched pairs revealed that knowledge about pharmacological methods (M_rank_ = 1.06; Me = 5.00), as compared to non-pharmacological methods (M_rank_ = 1.94; Me = 10.00), was statistically significantly lower, z = 17.822; *p* < 0.001; r_c_ = 0.85. The effect size was very high. The level of respondent knowledge was significantly higher in cases of non-pharmacological methods.

### 3.3. Knowledge about Pain Relief in Labor and Sociodemographic Variables

Spearman’s correlation was used to assess the correlation between the extent of knowledge about pain relief in labor and selected sociodemographic variables (Table 4).

The analysis revealed a weak correlation between the level of education and the level of knowledge about pharmacological (r_s_ = −0.13_,_
*p* = 0.007) and non-pharmacological (r_s_ = 0.14, *p* = 0.003) methods of pain relief in labor. Higher level of education correlated with lower knowledge about pharmacological methods. There was no statistically significant correlation between education and basic knowledge. A weak and negative correlation (r_s_= −0.10_,_
*p* = 0.026) was found between age and knowledge about non-pharmacological methods of pain relief, but no correlation was observed between age and basic knowledge as well as knowledge about pharmacological methods. A weak correlation was observed between the place of residence and knowledge about non-pharmacological methods and pharmacological methods. Bigger towns are associated with higher knowledge about non-pharmacological and lower knowledge about pharmacological methods of pain relief.

### 3.4. Knowledge about Relieving Labor Pain and Variables Related to Delivery

Correlation analysis was used to test the relationships between delivery-related variables and knowledge score about pain relief in labor (Table 5).

Parity, number of vaginal deliveries, number of cesarean sections, and hospital referral level of the previous childbirth did not correlate with the scores of basic knowledge and knowledge about non-pharmacological and pharmacological methods of pain relief in labor. The correlations presented in Table 5 were weak or very weak.

An additional analysis was conducted to test the knowledge of the respondents about pain relief in labor versus history of delivery in hospitals with different referral levels (primary, secondary, and tertiary). The Kruskal–Wallis test was used to analyze the differences between the three groups (Table 6). No significant differences between women who gave birth at the primary, secondary, and tertiary referral center and the level of their knowledge about pain relief in labor were found. The effect sizes were small (Table 6).

The level of knowledge about pain relief in labor versus the mode of delivery (subdivided into only vaginal delivery, only cesarean delivery, and both) was analyzed using the Kruskal–Wallis test (Table 7). Depending on the mode of delivery, significant differences in the knowledge about non-pharmacological and pharmacological methods were found between the groups. However, the effect size was small for both analyses.

Post hoc analysis revealed that women with a history of only vaginal or only cesarean delivery had statistically significantly higher scores as far as knowledge about non-pharmacological and pharmacological methods of pain relief in labor was concerned, as compared to those with a history of both delivery modes. The differences were not statistically significant, either for non-pharmacological or pharmacological methods of pain relief in labor, among women who had vaginal delivery or cesarean section. The extent of basic knowledge proved to be the same among the women with history of different modes of delivery.

## 4. Discussion

Apart from the physiological factors, the course of labor depends on the competence of the medical personnel, not to mention patient knowledge about the stages of labor and pain-relief methods in childbirth. Patient knowledge about different stages of labor and their duration results in better cooperation with the medical personnel and improves the feeling of safety in the parturients. However, the number of the available sources about the link between problems with controlling labor pain and the lack of reliable knowledge in the parturients is extremely limited.

In this study, we found that antenatal classes and the Internet were the most frequently reported sources of knowledge (68.9% and 54.6%, respectively). Interestingly, the Internet as a knowledge source was listed more often than a midwife, a physician, or professional literature. The opinions of friends (19.2%) were also mentioned more frequently as a knowledge source than a physician. Our findings are consistent with other reports, which also demonstrated that the Internet served as the chief source of knowledge about the normal course of labor and methods of pain relief in labor for 63% up to 80% of the respondents [2,17,20]. Most respondents (22.3%) obtained knowledge about non-pharmacological methods of labor pain relief from websites, blogs, Internet forums, webpages and groups on social networking sites run by midwives or obstetricians, and only 14% from midwives, and 5% from obstetricians [21]. Australian studies demonstrated that antenatal classes, multimedia, and friends/relatives were the most popular sources of knowledge about pain relief in labor [22,23]. Similar findings were reported by Aggel et al. in a population of women in Saudi Arabia, where medical personnel, friends, and family constituted the main sources of knowledge about pain relief in labor, although their results were in contrast with our observations [24].

A study among 60 primiparas (aged 20–30 years) investigated whether prenatal education affected the course of pregnancy, labor, and early motherhood and found that 60% of the participants of antenatal classes used the knowledge obtained from a midwife. In turn, the primiparas who did not participate in antenatal classes relied on knowledge from the Internet (90%) and professional literature (83.3%) as compared to 33.3% of the antenatal class participants [25]. The effectiveness of prenatal education has been confirmed by numerous studies [21,22,25,26]. Only 3% of the participants of antenatal classes had no knowledge about the methods of relieving labor pain as compared to 25% of women who received no prenatal education [27]. The percentage of women who participate in antenatal classes remains very small. According to Pilewska-Kozak et al., only 35.7% of the parturients took part in antenatal classes [21]. The literature demonstrates that women who attend prenatal education classes, which cover the topic of pain management in childbirth, report their labor pain to be statistically significantly lower [28]. The analysis of our findings and of the available literature demonstrated that in order to improve women’s knowledge about both pain-relief methods and the course of labor, it is vital to popularize antenatal classes so the parturients may obtain their knowledge from reliable sources. Moreover, it is important to increase the knowledge of midwives about ways of relieving pain in childbirth. According to various studies, most healthcare providers are aware of various approaches to pain management including both pharmacological and non-pharmacological options. However, half of all healthcare providers consider labor pain as ‘natural’ and necessary for birth, and therefore do not routinely provide pharmacological pain relief [29]. This has been also confirmed by studies that demonstrated that the overall use of labor pain-relief methods was 34.4% (30.4% non-pharmacological and 8.4% pharmacological), with more than half of the obstetric caregivers (54.2%) having adequate knowledge about labor pain-relief methods [30].

The overall use of labor pain-relief methods by obstetric caregivers was low. When it comes to pharmacological methods of relieving labor pain, the most common methods reported by the respondents were the use of Diclofenac (153/51.2%), Paracetamol (140/46.8%), Pethidine (102/34.1%), and Hyoscine (93/31.1%). Furthermore, 29.1% of the study participants reported that the presence of a birth partner during labor was not allowed by their health center, and 87% of them reported that they received no special training about labor pain management. A statistically significant relationship was found between the knowledge, attitude, and work experience of the healthcare providers and the use of pain-relief methods in labor [30]. A literature review found that a lack of knowledge and obtaining it from social media was one of the barriers to using non-pharmacological methods to relieve labor pain [31].

Childbirth is indispensably connected with the topic of labor pain and ways of relieving that pain. All pregnant women fear labor pain. Knowledge about methods of pain relief in labor affects the frequency of their application. The literature offers information about several well-known methods of relieving labor pain; however, global studies emphasize insufficient knowledge about non-pharmacological methods of pain relief in childbirth among women [10,32,33,34]. According to a study conducted in Nigeria, among 245 women, only 68.6% of the women possessed knowledge about non-pharmacological ways of relieving pain in labor, but most of them perceived their knowledge to be too insignificant to apply those methods in practice. The most well-known methods included breathing techniques (51.8%), massage (36.7%), changes in position (32.2%), and relaxation techniques (26.5%) [10]. These values are very different from our findings, which were as follows: massage—89.2%; breathing techniques—88,1%; upright birthing positions—75.6%; immersion in water—71.4%; physical activity—63.9%. Similar data were reported by a Brazilian study among 165 women, where almost all participants (96.5%) had knowledge about at least one non-pharmacological method of relieving pain in labor, with immersion in water (87.1%), the use of a birthing ball (80.7%), and breathing technique (74.8%) as the most common methods of pain management [2]. Similar findings have been reported by Finnish authors, who reported that the parturients were aware of the following non-pharmacological ways of relieving pain in labor: breathing technique, use of cold and hot compresses, physical activity, and upright positions [35,36].

Our results are different from studies conducted in India, which confirmed unsatisfactory levels of knowledge about non-pharmacological methods of pain management in labor [18,33], and from a Polish study by Król et al., who found that only a small percentage of women reported awareness of the following methods: breathing technique (17.0%), massage (14.9%), immersion in water (11.1%), and regional anesthesia (13.5%) [37]. The differences might be the result of limited access to knowledge, personal beliefs, and preferences of the healthcare providers regarding labor pain, which have been reported in the literature [29]. Additionally, more than a decade has passed since these studies were conducted. Therefore, the wide access to various sources of information may have had an impact on the better knowledge of our respondents and influenced the results of our study.

In this study, at least half of the respondents scored the maximum number of points in the ‘basic knowledge’ area. The analysis of the knowledge score revealed that knowledge about the pharmacological methods of pain management in labor (M_rank_ = 1.06; Me = 5.00) as compared to the non-pharmacological methods (M_rank_ = 1.94; Me = 10.00) was statistically significantly lower (*p* < 0.001; r_c_ = 0.85). The level of knowledge among the study population was significantly higher in cases of non-pharmacological methods and was deemed ‘satisfactory’. Our findings are in contrast to the global reports. According to the literature worldwide, the level of women’s knowledge about non-pharmacological [10,18,38,39] and pharmacological [18,32,40,41,42] methods of pain management strategies in labor remains low. Approximately 16.3%–35.3% of the respondents report awareness of at least one technique of non-pharmacological pain relief in labor [10,18,38,39]. Likewise, their knowledge about the effectiveness and benefits of pharmacological methods also leaves much to be desired [18,24,43,44]. Ibach et al. investigated the knowledge and expectations regarding labor pain among primigravidas and found that many women expected the labor pain to be excruciating, but none of them knew about the negative consequences of labor pain and why it was necessary to use pain-relief methods. According to those authors, the study participants had limited knowledge about pain-management strategies but a significant number of the respondents declared they would want to expand their knowledge about the topic in question, which is important. Those authors concluded that primigravidas lacked adequate prenatal education [45], which is in contrast to our findings, as there was no relationship between parity and the level of knowledge among our respondents. Moreover, Heim et al. confirmed that there were no significant differences between nulliparous and parous women as far as knowledge about non-pharmacological techniques for pain relief during childbirth was concerned [2].

Sufficient levels of knowledge and acceptance of labor pain were reported in a cross-sectional study by Christiaens et al. among women from Holland and Belgium [46], which is consistent with our findings. Additionally, those authors emphasized that labor pain acceptance and personal control in pain relief are important in coping with labor pain [46]. Another Polish study confirmed the high percentage of women expecting anesthesia for childbirth over the past decade. There was no difference between 2010 and 2020 in the percentage of woman who wanted analgesia during labor, 67.9% and 73.9%, respectively [47]. Notably, a study conducted in Australia demonstrated a large discrepancy between perception and actual knowledge about analgesia during labor. Therefore, healthcare providers need to be aware that women overestimate their knowledge and understanding of the analgesic options [23]. In this study, we found that the respondents possessed knowledge about the benefits of using pain relief during labor, both non-pharmacological and pharmacological. In the study conducted by Babiker et al. 61.8% of respondents had correct knowledge of epidurals [48]. Still, a systematic review of the literature by Thomson et al. revealed that women need information about risks and benefits associated with all available pain-relief methods [49].

In this study, we demonstrated a weak and negative correlation (r_s_ = −0.13_,_
*p* = 0.007) between the level of education and knowledge about pharmacological methods and a weak and positive correlation (r_s_ = 0.14, *p* = 0.003) between education and knowledge about non-pharmacological methods. Therefore, it seems safe to conclude that higher level of education corresponds to higher knowledge about non-pharmacological but lower knowledge about pharmacological ways of relieving labor pain. The correlation between higher level of education and knowledge about pain-relief methods in childbirth has been well-documented by numerous studies conducted in Nigeria, Australia, Saudi Arabia, or Finland [22,35,44,50].

In this study, a weak and negative correlation was found between age and knowledge about non-pharmacological methods, and statistically insignificant correlation between age and basic knowledge. A weak and positive correlation was detected between the place of residence and knowledge about non-pharmacological methods and a weak and negative correlation was found between the place of residence and basic knowledge. Interestingly, larger towns as the place of residence were linked with higher knowledge about non-pharmacological and lower knowledge about pharmacological methods of pain relief. Other sociodemographic variables such as parity, number of vaginal deliveries or cesarean sections, and hospital referral level did not correlate significantly with the level of basic knowledge and that about non-pharmacological and pharmacological methods. Our findings are in contrast with reports from many other countries, chief among them Ethiopia, India, Pakistan, Turkey, and Nepal, which demonstrated that previous childbirth experiences, including cesarean section delivery, have a positive effect on women’s knowledge about pharmacological methods of pain relief in labor [51]. Notably, in developing countries, the availability and knowledge about pharmacological methods of pain relief depended on the place of residence, with higher knowledge about the methods in question among inhabitants of urban areas [24,39].

Śledzińska et al. demonstrated that knowledge about methods of pain relief in labor does not depend on age, parity, or place of residence, but it does depend on education and the increase is proportional to the increase in the level of education [52]. Miquelutti et al. reported that participants of antenatal classes had higher knowledge about non-pharmacological methods of pain management as compared to their peers who did not obtain prenatal education [53]. Gałązka et al. found that higher knowledge corresponds to higher level of hospital referral where the delivery took place because the availability of pain-relief methods depends on the referral level, which is not consistent with our findings [54]. The correspondence between higher knowledge and higher level of hospital referral has also been confirmed by other studies [2,55].

## 5. Limitations

Our study is not without limitations, chief among them low sociodemographic diversity in the study population. Women with higher education constituted 86.6% of the sample, which made it impossible to assess the effect of education on the level of knowledge in question. It cannot be excluded that the level of education had a significant impact on the high level of knowledge about the methods of pain relief in labor. In this regard, the study population may not be representative of the general population. This limits the generalizability of the findings and warrants caution in drawing broad conclusions.

Another limitation of our study is the use of a non-validated instrument. However, we presented our survey tool in the manuscript to enable clear understanding and reproducibility of the research. Additionally, the data collected in the study were self-reported; therefore, we cannot exclude social desirability or recall bias. However, since the survey was anonymous, this may be a limited factor. The online survey also ensured that there was no influence from the researchers. The study used convenience sampling method as opposed to random sampling. Willingness to participate in the online survey determined the study group.

## 6. Conclusions

In this study, we found that antenatal classes and the Internet are the most popular sources of information for women. The most common non-pharmacological methods of pain relief in labor included: massage, breathing techniques, upright position, immersion in water and physical activity, whereas the most often listed pharmacological methods included epidural anesthesia and nitrous oxide. Almost all respondents found pain relief in labor to be beneficial. The knowledge of the respondents about pharmacological methods was statistically significantly lower as compared to non-pharmacological methods of pain relief in labor.

## 7. Implications for Obstetric Practice

Our results are a valuable contribution to the level of women’s knowledge and their sources of information about labor pain. The results we obtained can initiate a debate on the need for health care providers to popularize prenatal education. Efforts should be made to disseminate reliable information, create information standards and make it mandatory for women to participate in prenatal classes or educational workshops. Unfortunately, the sources of knowledge that women use are not always reliable. Therefore, knowledge about methods of coping with labor pain should be disseminated by professionals. Our study highlighted the need for a strategy to educate pregnant and birthing women about methods of labor pain relief and their benefits for both mother and baby. A possible solution could be a central national database of information based on evidence-based medicine.

## Figures and Tables

**Figure 1 healthcare-11-01882-f001:**
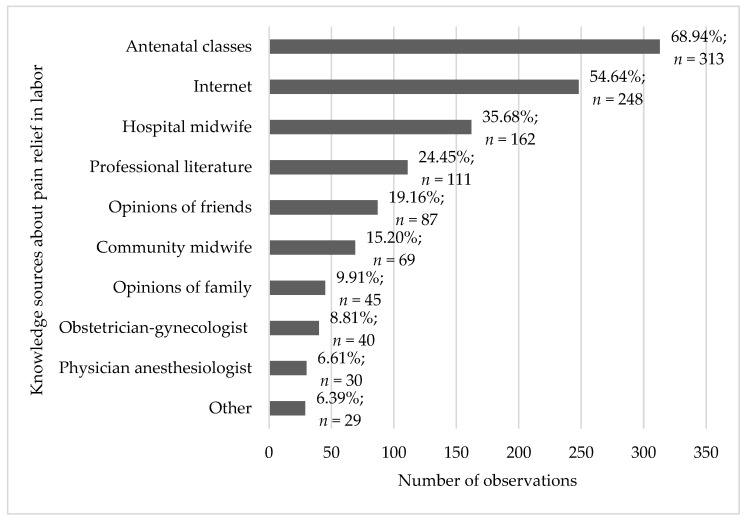
Frequency analysis of knowledge sources about pain relief in labor (*n* = 454).

**Figure 2 healthcare-11-01882-f002:**
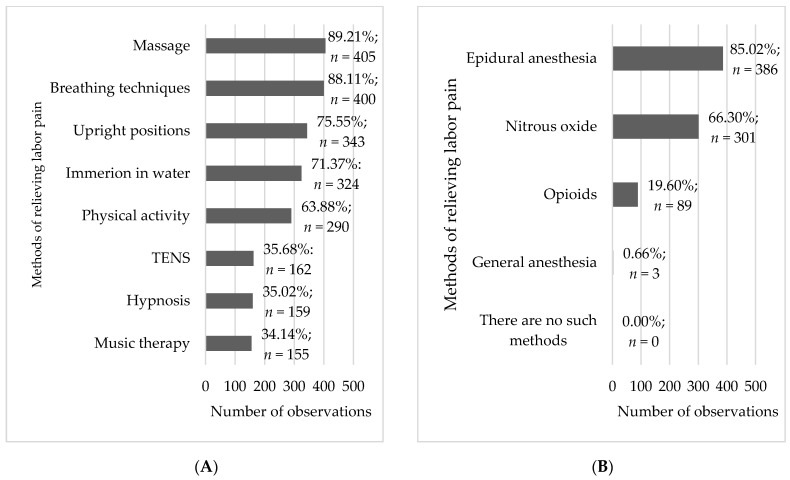
Frequency analysis of non-pharmacological (**A**) and pharmacological (**B**) methods of relieving labor pain (*n* = 454).

**Figure 3 healthcare-11-01882-f003:**
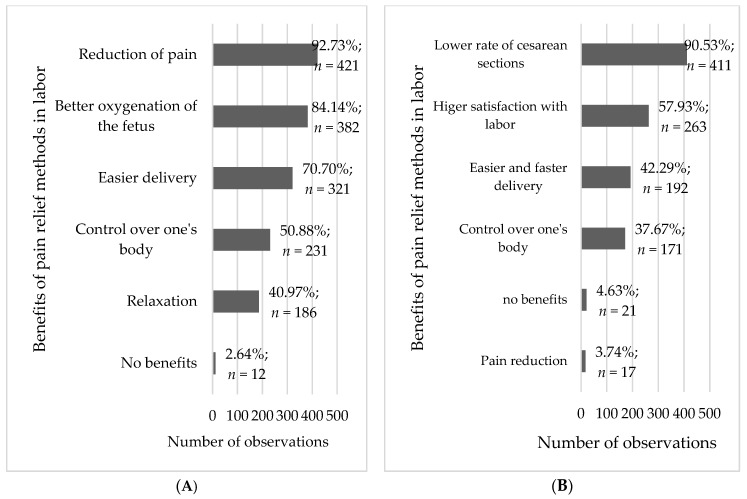
Frequency analysis of benefits of using non-pharmacological (**A**) and pharmacological (**B**) methods of pain relief in labor (*n* = 454).

**Table 1 healthcare-11-01882-t001:** Questions about pain relief in labor within the investigated areas of knowledge.

Knowledge about Non-Pharmacological Methods	Knowledge about Pharmacological Methods	Basic Knowledge
Which of the following belong to the non-pharmacological methods of pain relief in labor (select more than one choice, if appropriate)	Which of the following belong to the pharmacological methods of pain relief in labor (select more than one choice, if appropriate)	Which legal act regulates the usage of pain relief methods in labor
Can non-pharmacological methods have a negative impact on the neonate	Can pharmacological methods have a negative impact on the course of labor and neonatal condition after birth	Do you believe that labor pain is a physiological symptom
What are the benefits of appropriate breathing technique during labor (multiple-choice question)	Is epidural anesthesia associated with higher probability of cesarean section delivery?	Are there any methods of relieving labor pain
Do you know what upright birthing positions are	Do you believe patients in Poland are charged extra for pharmacological pain relief in labor	
Is the change of the birthing position to the upright one associated with lower labor pain	What are the benefits for the mother of using pain relief in labor (select more than one choice, if appropriate)	
Is it important to have the support of a birth partner during labor	Is the use of nitrous oxide associated with adverse events in the neonate	

**Table 2 healthcare-11-01882-t002:** Frequency analysis of the sociodemographic variables in the study group (*n* = 454).

Sociodemographic Variables		*n*	%
Age (in years)	18–23	6	1.3
	24–30	162	35.7
	31–35	217	47.8
	>35	69	15.2
Education	Primary	0	0.0
	Junior high school	1	0.2
	Vocational	55	12.1
	Secondary	5	1.1
	Higher	393	86.6
Place of residence	Rural area	59	13.0
	Town up to 100 thousand	67	14.8
	Town of 100–250 thousand	78	17.2
	Town over 250 thousand	250	55.0
Vaginal delivery	0	114	25.1
	1	248	54.6
	2	77	17.0
	3	10	2.2
	>4	5	1.1
Cesarean section	0	314	69.2
	1	114	25.1
	2	26	5.7
	3	0	0.0
	>4	0	0.0
Hospital referral level	Primary	156	34.4
	Secondary	143	31.5
	Tertiary	155	34.1

**Table 3 healthcare-11-01882-t003:** Distribution analysis of knowledge scores about methods of pain relief in labor.

Knowledge about Pain Relief in Labor	Min	Q1	Q2	Q3	Max	M	SD	SKE	K	K-S
Non-pharmacological	2.00	8.00	10.00	12.00	16.00	10.33	2.86	−0.12	−0.39	0.082 *
Pharmacological	2.00	4.00	5.00	6.00	10.00	5.18	1.60	0.16	−0.52	0.126 *
Basic knowledge	0.00	2.00	3.00	3.00	3.00	2.37	0.87	−1.10	0.01	0.366 *

Min—minimum; Q—quartile; Max—maximum; M—mean; SD—standard deviation; SKE—skewness; K—kurtosis; K-S—Kolmogorov–Smirnov test value * *p* < 0.01.

**Table 4 healthcare-11-01882-t004:** Spearman’s correlation between knowledge score about methods of pain relief in labor and sociodemographic variables.

Selected Sociodemographic Variables	Knowledge about Pain Relief in Labor					
	Non-Pharmacological		Pharmacological		Basic Knowledge	
	r_s_	*p*	r_s_	*p*	r_s_	*p*
Age	−0.10	0.026	−0.07	0.125	−0.08	0.074
Education	0.14	0.003	−0.13	0.007	0.08	0.107
Place of residence	0.09	0.056	−0.10	0.033	0.09	0.043

r_s_—Spearman’s correlation, *p*—significance.

**Table 5 healthcare-11-01882-t005:** Spearman’s correlation between knowledge about methods of pain relief in labor and delivery-related variables.

Variables Related to Delivery	Knowledge about Pain Relief in Labor					
	Non-Pharmacological		Pharmacological		Basic Knowledge	
	r_s_	*p*	r_s_	*p*	r_s_	*p*
Parity	−0.09	0.061	0.04	0.356	−0.03	0.580
Vaginal delivery	−0.04	0.393	−0.01	0.853	−0.02	0.647
Cesarean deliver	−0.04	0.449	0.01	0.808	0.01	0.911
Hospital referral level	0.00	0.975	−0.06	0.213	0.08	0.076

r_s_—Spearman’s correlation; *p*—significance.

**Table 6 healthcare-11-01882-t006:** Summary of the analysis of Kruskal–Wallis difference test of knowledge among women giving birth in hospitals with different referral levels.

Knowledge about Pain Relief in Labor	Hospital Referral Level						H_(2)_	*p*	ε^2^
	Primary N = 156		Secondary N = 143		Tertiary N = 155				
	M_rank_	Me	M_rank_	Me	M_rank_	Me			
Non-pharmacological	219.32	10.00	245.89	11.00	218.77	10.00	4.148	0.126	0.01
Pharmacological	242.86	5.00	213.81	5.00	224.68	5.00	3.895	0.143	0.01
Basic knowledge	213.87	3.00	232.05	3.00	237.02	3.00	3.472	0.176	0.01

Note. Post hoc analyses were not performed due to insignificant level of Kruskal–Wallis test. N—number of observation; M_rank_—ranks of results; Me—median; H—result of Kruskal–Wallis’s test; *p*—significance; ε^2^—effect size.

**Table 7 healthcare-11-01882-t007:** Summary of the analysis of Kruskal–Wallis difference test of knowledge among women with history of different modes of delivery.

Knowledge about Pain Relief in Labor	Methods of Delivery						H_(2)_	*p*	ε^2^	Post Hoc
	Vaginal (1) N = 313		Cesarean Section (2) N = 113		Both Methods (3) N = 27					
	M_rank_	Me	M_rank_	Me	M_rank_	Me				
Non-pharmacological	230.22	10.00	234.22	11.00	159.43	9.00	7.815	0.020	0.02	1, 2 > 3
Pharmacological	226.04	5.00	244.77	5.00	163.78	4.00	8.681	0.013	0.02	1, 2 > 3
Basic knowledge	226.53	3.00	235.03	3.00	198.91	3.00	2.170	0.338	0.00	–

Note. Post hoc analyses were performed with multiple comparisons method. N—number of observation; M_rank_—ranks of results; Me—median; H—result of Kruskal–Wallis’s test; *p*—significance; ε^2^—effect size.

## Data Availability

Data is contained within the article.

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
