# Peer review of "Women’s Knowledge about Pharmacological and Non-Pharmacological Methods of Pain Relief in Labor"

_healthcare, 2023, doi:10.3390/healthcare11131882_

Round 1
Reviewer 1 Report
The paper has been improved, and I am happy to accept the paper.
Author Response
Thank you very much.
Reviewer 2 Report
Dear Authors
Thank you for your efforts. The changes you have made have improved the manuscript. For my part it can be published, but you should introduce as a limitation of your study the use of a non-validated instrument. Otherwise I believe that the changes introduced improve the final result. Best regards
Author Response
Dear Reviewer,
Thanks very much for your positive review and your acceptance of the changes so far. We found the reviewer comments very helpful, thus we revised the paper accordingly. Please find attached a revised version of our manuscript, which we would like to resubmit for publication.
We hope, that after the last changes that we have made the manuscript will meet with your approval.
Another limitation of our study is the use of a non-validated instrumental. However, we presented our survey tool in the manuscript to enable clear understanding and reproducibility of the research.
Kind regards,
Jakub Pietrzak
